# Bioactive Compounds and Adipocyte Browning Phenomenon

**Josué Manríquez-Núñez and Minerva Ramos-Gómez ***

Departamento de Investigación y Posgrado de Alimentos, Facultad de Química, Universidad Autónoma de Querétaro, Centro Universitario S/N, Cerro de las Campanas, Querétaro C.P. 76010, Mexico; josuemn.11@hotmail.com
* Correspondence: minervaramos9297@gmail.com

**Abstract:** Overweight and obesity have become worldwide health issues in most countries. Current strategies aimed to prevent or reduce overweight and obesity have mainly focused on the genes and molecular mechanisms that give the functional characteristics to different types of adipose tissue. The Browning phenomenon in adipocytes consists of phenotypic and metabolic changes within white adipose tissue (WAT) activated by thermogenic mechanisms similar to that occurring in brown adipose tissue (BAT); this phenomenon has assumed great relevance due to its therapeutic potential against overweight and obesity. In addition, the study of inflammation in the development of overweight and obesity has also been included as a relevant factor, such as the pro-inflammatory mechanisms promoted by M1-type macrophages in adipose tissue. Studies carried out in this area are mainly performed by using the 3T3-L1 pre-adipocyte cell line, testing different bioactive compound sources such as plants and foods; nevertheless, it is necessary to standardize protocols used in vitro as well to properly scale them to animal models and clinical tests in order to have a better understanding of the mechanisms involved in overweight and obesity.

**Keywords:** adipocyte browning; bioactive compounds; 3T3-L1 cell line

## 1. Introduction

The classic function of the adipose tissue is the storage of energy in the form of triglycerides; however, it is also an important signaling organ with endocrine and paracrine functions [1]. In individuals with normal weight, adipose tissue represents between 10 to 30% of body weight [2]; however, in subjects suffering from morbid obesity, it can be up to 80% of individual weight [3].

Adipose tissue is currently classified into: white adipose tissue (WAT), brown adipose tissue (BAT) and beige adipose tissue (BeAT). This classification was established, at first, based on the characteristic color of the different tissues and later by the molecular markers that each tissue expresses, as well as the mitochondrial properties (thermogenesis). In this regard, BAT is associated with the expression of genes related to thermogenesis, mainly the uncoupling protein 1 (UCP-1), and it has become a critical point in obesity research due to adaptive thermogenesis, which is the process of heat production regulated in part by the catabolism of substrates without the release of energy by ATP; while WAT has the main function of storing lipids. For its part, BeAT has the heat-generation characteristics of BAT; but interestingly, this tissue emerges in the WAT regions [4,5]; this has an important impact on the adipose-tissue investigation towards overweight and obesity control strategies. Major differences among adipose-tissue types are summarized in Table 1.

Despite the complexity of the development in these health issues, the main strategies in order to prevent and control the progression of overweight and obesity conditions mainly relay on physical activity and changes in diet and nutritional patterns [6,7]. In recent years, comprehensive research on how various bioactive compounds (pure or extracts) in the diet modulate the mechanisms related to the development in overweight and obesity, has been one of the most innovating strategies to reduce the impact on the health

sector and the general population due to the increase in the incidence of overweight and obesity worldwide.

**Table 1.** Characteristics and differences among white (WAT), brown (TAB) and beige (BeAT) adipose tissues.

| | WAT | BAT | BeAT | Authors |
|---|---|---|---|---|
| Anatomical location | Subcutaneous and visceral | Adrenal, interscapular, and neck area in human infants | WAT deposits and supraclavicular region | [4,8] |
| Morphology | Large adipocytes | Small adipocytes | Small adipocytes | [9,10] |
| Lipid droplets | Large | Multiple and small | Multiple and small | [8–10] |
| Origin/development | Progenitor Pdgfr-$\alpha$ | Progenitor Myf5$^+$ | Progenitor Pdgfr-$\alpha^+$ | [4,8] |
| Primary function | Energy storing | Heat production | Heat production | [11] |
| Endocrine signals/adopokines | Adiponectin, Adipsin, Omentin, IL-4, IL-6, IL-10, Leptine, Resistin, Visfatin, Chemerin, TNF-$\alpha$, MCP-1 | FGF-21, NRG-4, Myostatin, IGF-1, CXCL-14, BMP-8b, VEGF-A, T3, IL-6, GDF-15, Adiponectin, S100b, NGF and EPDR1 | IGFBP-2, METRNL, IL-6, GDF-15 and SLIT2-C | [9,12] |
| Mitochondrial UCP1 | Low/high upon stimulation | Low/high upon stimulation | Low/high upon stimulation | [13] |
| Expressed genes | Asc1, Fabp4, Fbxo31, Leptin, Lpl, Mpzl2, Nr1H3, Nrip1, Rb1, Rbl1, Resis-tin, Serpina3K, Tcf21 and Wdnm1 | Bmp7, Efb2, Ednrb, Eva1, Mir133B, Mir206, Myf5, Pdk4, Prex1 and Zic1 | Aqp7, Asc1, Car4, Cd137, Cd40, Cited1, Ear2, Shox2, Slc27A1, Sp100, Tbx1 and Tmem26 | [4] |
| Mitochondrial biogenesis | Low | High | Medium | [14] |
| Activated hyperplasia or hypertrophy | In prolonged positive energy balance conditions, adipocytes expand cell size (hypertrophy) and number (hyperplasia). | An increase in thermogenic activity derived from physical activity has been reported; however, it has not been related to tissue expansion. | Not reported | [15–17] |
| Insulin resistance | Led by sustained low-grade inflammatory process. | Negative | Negative | [14,15,17] |

## 2. Adipocyte Browning Phenomenon

Adipose tissue is the main form of energy storage, whose purpose is to guarantee the survival of the major functions of the body during prolonged periods of fasting. Despite this important role, easy access to different sources of hypercaloric foods has led to an epidemic of overweight and obesity worldwide [18]. Initially, it was believed that both WAT and BAT come from the same mesenchymal progenitor cells, because, apparently, both tissues require PPAR$\gamma$ for their development; however, currently it is known that some BAT depots arise from Myf5$^+$ myogenic precursor cells and the induction of the PRMD16 gene [19]. Although the Myf5$^-$ myogenic precursor is the most common progenitor accepted for WAT development, due to the plasticity that white adipocytes show, current research continues to identify those cells given origin to WAT deposits [20]. In mice, beige adipocytes do not derive from the same precursors that give origin to brown adipocytes (Myf5$^+$ precursor); meanwhile, in humans, BAT regions are abundant in infancy and decrease when the individual advances into adulthood [21]. BAT has its development in the embryonic phase while the BeAT can be induced in WAT regions (postnatal life) [22].

The browning phenomenon process is, in part, driven by the UCP1 protein that anchors itself to the inner mitochondrial membrane and, when present, uncouples ATP production derived from lipid and carbohydrate catabolic pathways; therefore, the energy produced is released in the form of heat [18]. Another significant difference is that brown adipocytes express high levels of UCP-1 in basal conditions, whereas beige adipocytes only express this protein apparently in response to PPAR$\gamma$ or ADRB agonists [23,24]. In this sense, the mechanisms by which adaptive thermogenesis in beige adipocytes is activated have become the focus of several working groups.

The expression of UCP1 in white adipocytes is practically undetectable; interestingly, its induction can be triggered by dietary bioactive compounds; therefore, this stimulation could represent an explanation for the browning phenomenon on WAT. Moreover, these beige adipocytes share several phenotypic and metabolic characteristics with brown adipocytes, such as high levels of UCP1, activation of mitochondrial biogenesis and nuclear factors that increase the transcription of thermogenic proteins (mainly PPAR$\gamma$, PGC-1$\alpha$ and PRDM16; Table 1) [4,25].

Recent studies have reported that increased fat utilization through thermogenic activity promotes browning in WAT and attenuates obesity-induced inflammation and insulin resistance by the modulation of tissue-resident macrophages [26].

### 3. In-Vitro Studies of Adipocyte Browning

As mentioned before, several working groups have focused their efforts on identifying underlying mechanisms modulated by natural compounds (bioactive compounds), mostly due to the fact that in-vitro studies have shown the considerable potential of the induction of the adipocyte browning process. The vast majority of the studies have evaluated the effect of pure bioactive compounds present in some medicinal plants and several food sources, from spices to fruits or vegetables and their extracts (Table 2). In this regard, the first studies that were carried out to evaluate the mechanisms of adipocyte browning were performed with herbal-extract supplementation.

Ginsenosides are triterpenoid saponins present almost exclusively in the Panax plant genus (ginseng). In general, ginsenosides show a wide range of pharmacology effects on the central nervous, cardiovascular and endocrine systems in animal models, as well as in cell cultures [27]. Compound K, a ginsenoside from intestinal bacterial metabolism, was one of the first compounds to demonstrate its effect on the browning phenomenon process; this bioactive compound modulates white adipocyte activity by decreasing adipogenesis and, as a consequence, lipid accumulation [28]. Shortly after, in 2015, the effects of another ginsenoside, Rb1 ginsenoside, on adipocyte browning were explored in a pre-adipocyte cell-culture model and the results showed that Rb1 ginsenoside stimulates a significant increase in basal glucose uptake. Interestingly, Rb1 ginsenoside also significantly increased the relative expression of thermogenic genes such as PGC-1$\alpha$, PRMD16 and UCP-1 [29].

Later, in 2017, Jeong et al. supplemented albiflorin, which is a natural compound from *Paeonia lactiflorato*, to human amniotic mesenchymal stromal cells (hAMSCs) differentiated into white adipocytes, finding a significant decrease in lipid accumulation and adipogenic gene expression [30]; likewise, Kim et al. also conducted a study with hAMSCs cell line in which they evaluated the effect of farnesol, a terpene commonly found in essential oils from different plant sources, also reporting a decrease in lipid accumulation as well as decreased activity of PPAR$\gamma$ and increase in UCP-1 [31]. In 2019, using this same hAMSc cell line, the effect of $\beta$-lapachone, a compound which is obtained from the bark of the lapacho tree, was evaluated, finding similar results on the activation of thermogenic genes as well as a decrease in lipid accumulation [32]. In the same year, Velickovic et al. suggested a differential induction of thermogenic genes by caffeine in two mesenchymal cell lines, mMSCs and hMSCs, from mice and humans, respectively; however, results were obtained concerning the induction of UCP-1 and PGC-1$\alpha$, as well as an increase in oxygen consumption, which is also a characteristic of this thermogenic activity [33]. In-vitro studies conducted in different cell lines suggest that the induction of browning mechanisms can be observed in a similar way despite being different organisms, which indicates a high similarity in the elements involved in this process.

Recently, in 2020, the potential effect of Rb3 ginsenoside on the browning phenomenon was also evaluated, and the results indicated that this compound can promote browning from white differentiated adipocytes to beige adipocytes partially due to the increase in AMPK activity [34]. As shown by these studies, different members of the ginsenoside compounds exert a positive implication over the browning process; nevertheless, only in recent years have efforts been directed in order to identify more clearly the signaling pathways and molecular elements involved in this phenomenon.

Other compounds from plants used mostly for therapeutic purposes in traditional medicine have also been evaluated to promote adipocyte browning. Magnolol, classified as a lignin isolated from *Magnolia officinalis*, is widely used for its therapeutic properties such as the alleviation of pain and anxiety and also shows a normalization of cardiovascular disease risk in humans [35]. More importantly, it has also been shown to effectively reduce lipid accumulation in obese mouse models [36]. In cell cultures, magnolol up-regulated several genes related to the browning process during pre-adipocytes differentiation to beige adipocytes such as PGC-1$\alpha$, PRMD16, CD137 and Tbx1, among others [37]. This effect is apparently also promoted by AMPK activity and PPAR$\gamma$/Sirt1, which leads to an increase in the transcription rate of the thermogenic genes and energy expenditure.

Although there is mounting evidence about the effects of bioactive compounds from medicinal plant and herbal sources, recently, research has focused on the study of bioactive compounds from food ingredients as well as fruits and vegetables on the browning process; this is because part of the current strategy to prevent overweight and obesity worldwide is focused on the improvement in the nutritional content components in foods as well as the inclusion of certain food groups into the diet, with proven beneficial effects on these health conditions. Bioactive compounds such as δ-tocopherol, trans-anethole and quercetin can be found in different spices or ingredients used in food production. δ-Tocopherol can be found in grain oils, seeds and some vegetables such as spinach and broccoli; this type of bioactive compound has been related to an increase in PPARγ activity [38]. In differentiated white adipocytes, δ-tocopherol has been identified as an agonist of PPARγ and, in this sense, as a promoter of the transcription of thermogenic genes [39]. Similarly, bioactive compounds such as trans-anethole (mostly found in anise) and thymol (an important component of thyme) have shown induction of the transcription rate of thermogenic genes including PGC-1α, PPARγ, PRMD16 and UCP-1 [40,41], reinforcing the idea that this phenomenon could be mainly driven by signaling pathways regulated by AMPK.

Compounds such as cianidine-3-glucoside (a polyphenol compound from berries and grapes), gallotannin (hydrolyzable tannin from mango), lycopene (high content in tomatoes and some fruits) and resveratrol (mainly in grapes and nuts) have also shown the potential for regulating PGC-1α, PPARγ, PRMD16 and UCP-1 at the transcriptional and posttranslational level [42–45].

Most recently, these same thermogenic genes (PGC-1α, PPARγ, PRMD16 and UCP-1) can also be transcriptionally induced by complex food matrices such as grape pomace, raspberry extract and strawberry extract [46–48]. Moreover, some effects of these food matrices on the phenomenon of browning have yielded interesting results. Raspberry extract improves the mitochondrial biogenesis and the transcription rate of thermogenic genes (required events during the browning process); but it also decreases the signal of autophagy processes, which are also expected in a thermogenic adipocyte phenotype [48]. Similarly, strawberry extract modulates the activity of PPARγ, which has been proposed as a positive regulator of the browning process due to its participation in energy homeostasis [46].

**Table 2.** In-vitro effects of several bioactive compounds on the browning phenomenon.

| Compound/Extract | Concentration Range Evaluated | Results | Authors |
|---|---|---|---|
| Compound K (bacterial gingenoside) | 5 μM | Inhibits adipocyte maturation from pre-adipocyte to white adipocyte, decreases adipogenesis and lipid accumulation. | [28] |
| Ginsenoside Rb1 | 0.01, 0.1, 1, 10 and 100 μM | Improves glucose intake and induction of thermogenic genes involved in browning process. | [29] |
| Albiflorin | 10 and 20 μ | Decreases lipid accumulation and reduces adipogenic-related gene expression | [30] |
| Farmesol | 0.5 and 2 μM | Decreases lipid accumulation and adipogenic-related genes, and induces thermogenic activity | [31] |
| β-Lapachone | 0.5, 1 and 2 μM | Decreases lipid accumulation and induces thermogenic genes involved in browning process. | [32] |
| Caffeine | 1 μM | Increases oxygen consumption and thermogenic genes. | [33] |
| Ginsenoside Rg3 | 20 and 40 μM | Induction of thermogenic genes in mature differentiated white adipocytes. | [34] |
| Magnolol | 1, 5, 10 and 20 μM | Induction of thermogenic genes involved in browning process during pre-adipocyte maturation process. | [37] |
| δ-Tocopherol | 10, 50 and 100 μM | Induction of thermogenic genes involved in browning process in mature differentiated white adipocytes. | [39] |
| Thymol | 20 μM | Promotes mitochondrial biogenesis and increases lipid oxidation. | [40] |
| Trans-anethole | 1, 10, 50 and 100 μM | Decreases adipogenesis and lipogenesis during pre-adipocyte maturation process | [41] |
| Cyanidin-3-glucoside | 50 and 100 μM | Increases multilocular lipid droplets and mitochondrial biogenesis. | [44] |

**Table 2.** *Cont.*

| Compound/Extract | Concentration Range Evaluated | Results | Authors |
|---|---|---|---|
| Gallotannins | 2.5, 5, 10 and 20 mg/mL | Induction of thermogenic genes involved in browning process in pre-adipocytes and during pre-adipocyte maturation process. | [42] |
| Resveratrol | 10, 20 and 40 μM | Induction of thermogenic genes and decrease in lipid accumulation in mature differentiated white adipocytes. | [43] |
| Lycopene | 1, 2, 4 and 10 μM | Induction of thermogenic genes and decrease the lipid accumulation. | [45] |
| Grape pomace | 30 μM | Increases the β-adrenergic pathway and mitochondrial biogenesis. | [47] |
| Raspberry ketone | 50 and 100 μM | Induction of thermogenic genes and mitochondrial biogenesis. | [48] |
| Strawberry extract (*Fragaria x ananassa*) | 0, 10, 50 and 100 μg/mL | Inhibits adipocyte maturation from pre-adipocyte to white adipocyte. | [46] |

## 4. Adipocyte Mono-Culture 3T3-L1 and hAMSc Cell Line Studies

In the evaluation of the adipocyte browning phenomenon, one of the most widely used cell lines is the 3T3-L1 pre-adipocyte and the studies carried out using this pre-adipocyte cell line are shown in Table 3. According to the information, the initial growth condition was the standard indicated by ATCC, which includes incubation at 37 °C and 5% $CO_2$ under humidified atmosphere. Once the pre-adipocytes reach 80 to 100% of confluence (time 0) the different maturation media were applied (differentiation period) on cell cultures in order to mature the pre-adipocytes into white adipocytes. In addition, a few similar studies have been carried out on other cell lines, such as hAMSC, under comparable culture conditions, which are also detailed in Table 3.

**Table 3.** Composition of differentiation media and maturation period of different studies related to adipocyte browning performed on the 3T3-L1 and hAMSc cell lines.

| Compound/Extract | Differentiation Media | Differentiation Period | Bioactive Compound Incubation | Authors |
|---|---|---|---|---|
| Compound K (bacterial gingenoside) | DMEM, 10% FBS, 1% streptomycin-penicillin, 1 μg/mL Ins, 1 μM DEX, and 0.5 mM IBMX. | 8 days | 48 h after confluence. | [28] |
| Ginsenoside Rb1 | DMEM, 10% FBS, 0.1% Gentamicin, 0.05% Biotin, 1 μM Ins, 0.25 μM DEX, and 0.25 mM IBMX. | 7 up to 9 days | 1 h after differentiation period under starving conditions. | [29] |
| Cyanidin-3-glucoside | DMEM, 10% FBS, 1% streptomycin-penicillin, Ins, DEX and IBMX. | 7 days | During differentiation period. | [44] |
| Magnolol | DMEM, 10% FBS, 1% estreptomicina-penicilina, 10 μg/mL Ins, 0.25 μM DEX y 0.5 mM IBMX. | Not detailed | During differentiation period. | [37] |
| Quercetin | DMEM, 10% FBS, 1% streptomycin-penicillin, 1 μg/mL Ins, 0.25 mM DEX and 0.5 mM IBMX. | 11 days | During differentiation period (day 5 to 11). | [49] |
| Thymol | DMEM, 10% FBS, 1% streptomycin-penicillin, 10 μg/mL Ins, 0.25 μM DEX and 0.5 mM IBMX. | 6 up to 8 days | During differentiation period. | [40] |
| Gallotannins | DMEM, 10% FBS, 1% streptomycin-penicillin, 10 μg/mL Ins, 1 μM DEX and 0.5 mM IBMX. | 6 days | 48 h after differentiation period. | [42] |
| Grape pomace | DMEM, 10% FBS, 1% streptomycin-penicillin, 1 mM/L Ins, 0.25 mM/L DEX, 0.5 mM/L IBMX, and 0.1 mM/L indomethacin. | 10 days | 20 min after differentiation period. | [47] |
| Raspberry ketone | DMEM, 10% FBS, 1% streptomycin-penicillin, 10 μg/mL Ins, 0.25 μM DEX, and 0.5 mM IBMX. | 10 days | During differentiation period (day 6 to 8) | [48] |
| δ-Tocopherol | DMEM, 10% FBS, 1% streptomycin-penicillin, 10 μg/mL Ins, 2.5 μM DEX and 0.5 mM IBMX. | 10 days | During differentiation period. | [39] |
| Trans anethole | DMEM, 10% FBS, 1% streptomycin-penicillin, 10 μg/mL Ins, 0.25 μM DEX and 0.5 mM IBMX. | 6 up to 8 days | During differentiation period. | [41] |
| Ginsenoside Rg3 | DMEM, 10% FBS, 1% streptomycin-penicillin, 5 μg/mL Ins, 1 μM DEX and 0.5 mM IBMX. | Not detailed | 24 h after differentiation period. | [34] |

**Table 3.** *Cont.*

| Compound/Extract | Differentiation Media | Differentiation Period | Bioactive Compound Incubation | Authors |
|---|---|---|---|---|
| Lycopene | DMEM, 10% FBS, 1% streptomycin-penicillin, 10 µg/mL Ins, 1 µM DEX and 0.5 mM IBMX. | 6 up to 8 days | 24 and 48 h, respectively. | [45] |
| Resveratrol | DMEM, 10% FBS, 1% streptomycin-penicillin, 10 µg/mL Ins, 0.5 µM DEX and 0.5 mM IBMX. | 6 up to 8 days | During differentiation period. | [43] |
| Strawberry extract (*Fragaria x ananassa*) | DMEM, 10% FBS, 1% streptomycin-penicillin, 1 µg/mL Ins, 1 µM DEX, and 0.5 mM IBMX. | 10 days | During differentiation period. | [46] |

| Studies Performed in Other Cell Lines | | | | |
|---|---|---|---|---|
| **Compound/Extract** | **Cell Line** | **Differentiation Media** | **Differentiation Period** | **Bioactive Compound Incubation** | **Authors** |

| Compound/Extract | Cell Line | Differentiation Media | Differentiation Period | Bioactive Compound Incubation | Authors |
|---|---|---|---|---|---|
| Albiflorin | hAMSCs | DMEM, 10% SFB, 1% streptomycin-penicillin, 1 µg/mL Ins, 1 µM DEX, 0.5 mM IBMX and 100 µM Indomethacin | 14 days | After maturation, not detailed | [30] |
| Farnesol | hAMSCs | DMEM, 10% SFB, 100 U/mL streptomycin-penicillin, 1 µM Ins, 1 µM DEX, 0.5 mM IBMX and 100 µM Indomethaci | 15 days | During differentiation maturation (day 6 to 15) | [31] |
| β-Lapachone | hAMSc | DMEM, 10% SFB, 100 U/mL streptomycin-penicillin, 1 µM Ins, 1 µM DEX, 0.5 mM IBMX and 100 µM Indomethacin | 14 days | Not detailed | [32] |
| Caffeine | mMSCs/hMSCs | DMEM, 10% SFB, 100 U/mL strep-tomycin-penicillin, 10 µg/mL Ins, 1 µM DEX, 100 µM IBMX, 1 µM Rosiglitazone and 1 nM T3/DMEM, 10% SFB, 100 U/mL streptomycin-penicillin, 10 µg/mL Ins, 1 µM DEX, 500 µM IBMX, 1 µM Rosiglitazone and 1 nM T3 | Not detailed | 7 days | [33] |

The pre-adipose 3T3-L1 cell line was originally developed by clonal expansion from murine 3T3 cells [50]. Due to its potential to differentiate from pre-adipocytes (fibroblasts) to adipocytes, the cell line has been widely used for studies related to adipogenesis and general adipocyte biochemistry [51]. In this sense, it is also important to highlight that the components added into the maturation media for experiments with 3T3-L1 pre-adipocytes (DEX, IBMX and Ins) have been used since 1984 [52]. Back in 1984, the differences induced by these compounds in the transcription rate were investigated on the 3T3-L1 cell line; however, some of the genes related to adipocyte maturation were identified several years later.

To successfully differentiate 3T3-L1 cells from pre-adipocytes (fibroblast) to white adipocytes, pro-differentiative elements are usually added. The most commonly used agents include insulin (Ins), dexamethasone (DEX), and 3-isobutyl-1-methylxanthine (IBMX) at concentrations that usually ranged around 1 µg/mL, 0.25 µM, and 0.5 mM, respectively. Roughly 2 days after adding the differentiative agents, the cells should start to accumulate lipids (lipid droplets) that grow in number and size over time; this last distinctive morphology indicates that the pre-adipocytes have become white differentiated adipocytes [53].

Most studies performed with 3T3-L1 initiate the experimental protocol within 2 days of incubation in the presence of pro-differentiative elements; once this period is completed, the differentiation medium is changed for one consisting in DMEM, Ins and the bioactive compound at different concentrations for the rest of the differentiation period. Studies where the differentiation period is first concluded and the bioactive compound is subse-

quently administered in order to evaluate its effects on the browning process from white differentiated adipocytes are significantly fewer. It is important to note that, despite the fact that the pro-differentiative elements used as the basis for the differentiation medium are similar, the concentrations as well as the total differentiation period are distinct among working groups.

Mesenchymal cell lines such as hAMSCs, hMSCs and mMSCs, are cultivated under standard conditions of temperature, $CO_2$ and relative humidity. Furthermore, the composition of the medium used for their maturation into white adipocytes consists of the same elements supplemented to 3T3-L1 cell line, which includes Ins, IBMX and DEX; however, other elements such as rosiglitazone (Ros), triiodothyronine (T3) and indomethacin are added, whose main function is the induction of PPARγ [54,55].

## 5. Adipocyte Browning In Vivo and Clinical Studies

The study of the browning phenomenon in the 3T3-L1 pre-adipocyte cell line has provided clues about the potential molecular mechanisms related to it. However, to determine the contribution of these mechanisms for the development in new and innovating strategies in the prevention of overweight and obesity, it is necessary to explore these mechanisms in animal models as well as clinical studies in order to establish their efficacy in the prevention of these health conditions.

In mid 2014, Grimpo et al. performed an in-vivo study in order to evaluate the thermogenic activity of BAT by using UCP-1 knockout mice. Wild type and UCP-1 knockout animals were acclimated to temperatures of 5, 18 or 30 °C and subsequently evaluated using magnetic resonance. The findings showed a reorganization of BAT during cold acclimation (5 and 18 °C) towards an improvement in its thermogenic activity in the absence of UCP-1 (UCP-1 knockout mice) and no significant differences regarding wild-type mice were observed, which, in a novel way, shows that the thermogenic potential of BAT does not depend exclusively on UCP-1 [56]. In the same year, one of the first in-vivo studies aimed at monitoring the thermogenic activity driven by UCP-1 induction was carried out by using an in-vivo reporter system for endogenous UCP-1 expression. To achieve the goal, Galmozzi et al. [57] generated transgenic mice that expressed a luciferase signal under the control of UCP-1 locus. This represents a novel monitoring system for UCP-1 activity, mostly in BAT, to determine in vivo the thermogenic capacity in real time. Moreover, this strategy also opened up the possibility for generating an adipocyte cell line with the capacity of monitoring UCP-1 activity for in-vitro studies [57]. Likewise, in a complementary manner, Jeong et al. used a mouse model to evaluate the thermogenic effect of albiflorin and, by qPCR and Western blot analyses, they found that this compound increases the activity of UCP-1 and PGC-1α, which reinforces the results obtained in their study with the hAMSc cell line [30]. Performing a similar experimental scheme, Kim et al. observed that farnesol supplementation in mice also contributes to decreased weight gain as well as WAT browning, reporting an increase in thermogenic genes by using qPCR and Western blot [31]. In addition, Kwak et al. evaluate the effect of β-lapachone in a similar mouse model, which revealed decreased weight gain and increased thermogenic genes by qPCR and Western blot as well [32].

Consistently, Velickovic et al. conducted a study in young adults to whom they administered 65 mg of caffeine from a commercial brand in 200 mL of pure water. After 30 min, the increase in thermogenic activity in the supraclavicular region was determined by infrared spectrum with the help of thermally reflective skin markers. In addition, based on the results obtained from their cellular models, the authors concluded that caffeine may be an important biocompound that can potentially induce thermogenic activity in adults [33].

Moreover, the previous work carried out with the 3T3-L1 pre-adipocyte cell line helped to identify cellular and molecular elements involved in the browning process and thermogenic activity; however, only a few in-vivo and clinical studies have been designed to evaluate the effectiveness, similarities and orthologous genes involved in these mechanisms.

In 2017, Scotney et al. evaluated the effects of hydrocortisone in healthy subjects on the thermogenic activity of interscapular BAT using infrared thermography. The use of hydrocortisone was due to the fact that the endogenous elements that promote the BAT thermogenic activity in both humans and rodents have been poorly investigated; in part, for the technical limitations to assessing BAT activity in vivo [58]. Despite this, the study shows that endogenous elements such as cortisone can positively regulate the thermogenic activity of BAT in healthy individuals under normal conditions (without cold induction). They provided evidence that, in humans (in contrast to rodents), acute hypercortisolemia does not inhibit BAT function and despite that, the threshold of thermogenic activity cannot be overcome even during maximal short-term stimulation [58].

Continuing with the development of effective methods to monitor the thermogenic activity of adipose tissue, in 2018 Chan et al. designed a near-infrared fluorescent protein driven under UCP-1 promoter by viral transduction that works under adrenergic stimulation. Using multispectral optoacoustic imaging technology with ultrasound tomography, it was possible to detect, at 720 nm, the thermogenic activity in the adipose tissue where the vector had been implanted. Together, the system offers the possibility of being used for preclinical screening aimed to identify a variety of compounds that promote adipose tissue browning and thermogenic activity [59].

As mentioned before, one of the current and more effective strategies for the prevention of overweight and obesity worldwide is based on changes in dietary patterns. Phenolic compounds are widely distributed in fruits and vegetables with large beneficial effects on human health [60], and, more recently, their capacity to promote the browning phenomenon in adipose tissue. In this sense, in 2018, Lanzi et al. [47] evaluated the effects of a grape pomace extract on browning adipocyte induction in Wistar-Kyoto rats fed with a high-fat diet. After 10 weeks of a high-fat diet, the rats were supplemented with 300 mg/kg body weight/day of grape pomace extract and the results indicated that grape pomace extract increases the gene expression of the main transcriptional promoters of browning adipocyte phenomenon (PGC-1$\alpha$, PPAR$\gamma$, PRMD16 and UCP-1) in WAT. These findings support the hypothesis that the browning phenomenon could be driven in vivo by the consumption of phenolic compounds from fruits, vegetables and food by-products such as grape pomace [47]. All these studies are summarized in Table 4.

**Table 4.** Research about thermogenic characteristics and molecular elements related to the phenomenon of browning in in-vivo models and clinical studies.

| Model | Compound | Molecular Target | Technique | Authors |
|-------|----------|------------------|-----------|---------|
| Mouse | - | UCP-1 | Magnetic resonance detection | [8,56] |
| Mouse | Norepinephrine and rosiglitazone | UCP-1 | Transgenic mouse for UCP-1 luciferase signal | [57] |
| Mouse | Albiflorin | UCP-1, PGC-1$\alpha$, Nrf1, LIPIN1 and Glut4 | qPCR and Western blot analyses | [30] |
| Mouse | Farmesol | UCP-1 | qPCR and Western blot analyses | [31] |
| Mouse | $\beta$-Lapachone | UCP-1, PGC-1$\alpha$ and CIDEA. | qPCR and Western blot analyses | [32] |
| Human | Caffein | Supraclavicular region activity | Infrared thermography | [33] |
| Human | Hydrocortisone | - | Infrared thermography | [58] |
| Mouse | - | UCP-1 | UCP-1 fluorescent probe detected by ultrasonic tomography | [59] |
| Rat | Grape pomace | PPAR$\gamma$, MCP-1, PRDM16, PGC-1$\alpha$ | Primary adipocyte cell culture | [47] |

## 6. Macrophage Polarization on fat Browning by Dietary Compounds

Macrophages are an important effector element of the immune system; they are widely distributed across the body and exert a main role in a variety of processes, such as organ development, host defense, acute and chronic inflammation, tissue remodeling and homeostasis [61,62]. They can polarize depending on their surrounding microenvironment stimuli and signals; thus, polarized macrophages are mainly classified into two: classically activated macrophages or M1, which steer pro-inflammatory responses; and alternatively activated macrophages or M2, which are related to the activation of anti-inflammatory mechanisms [63].

Derived from the process of the accumulation of lipids within adipose tissue, it is possible to alternatively activate inflammatory-promoting mechanisms, which increase the mobilization and subsequent infiltration of monocytes into WAT [64]. Once infiltrated, these monocytes mature into macrophages, which, by recognizing pro-inflammatory stimuli, contribute to the maintenance of the inflammatory foci in the WAT. In the inflammatory focus, macrophages are activated to their M1 state, enhancing the release of pro-inflammatory cytokines by the activation of NF-κB [65].

Another scenario that may arise in healthy WAT or, in other words, in energy homeostasis, is that both infiltrated and resident macrophages in the tissue would tend to express anti-inflammatory cytokines such as IL-4 and IL-10 (type M2 macrophages). These cytokines limit the activity of pro-inflammatory processes, helping to maintain the functionality of WAT. In this sense, Mohammadi et al. supplemented curcumin to RAW 264.7 macrophages and observed a decrease in the activity of NF-kB, which is the main transcription factor related to M1 macrophages. These findings suggest the potential contribution of macrophage polarization in the activation of browning mechanisms in WAT [66] This type of research led to further studies assessing the interaction between macrophage polarization and WAT browning in co-culture models; for example, Mazur-Bialy and Poche evaluated the protective effect of riboflavin in a co-culture of RAW 264.7 and 3T3-L1, finding that this bioactive decreases the induction of pro-inflammatory cytokines such as TNFα, IL-6, and MCP-1, accompanied by an increase in anti-inflammatory elements such as adiponectin and IL-10 [67]. Soon after in 2019, Kang et al. evaluated the effect of brassinin in a co-culture system using the same RAW 264.7 macrophage and 3T3-L1 pre-adipocyte cell lines; interestingly, supplementation with this bioactive compound showed a reduction in the accumulation of lipids as well as the release of pro-inflammatory interleukins such as IL-6. According to their findings, they suggested that brassinin can inhibit obesity-induced inflammation in this co-culture model [68]. Subsequently, changes in the metabolic functions of WAT modify the type and amount of cytokines acting within tissue. Therefore, current research in the browning phenomenon resides in the importance of anti-inflammatory stimuli (endogenous and exogenous) and how these signaling molecules modulate the development of BAT/BeAT depots within adipose tissue against the progression of obesity and overweight [69].

## 7. Discussion

The research related the phenomenon of adipocyte browning has grown in relevance in recent years, due to rising health problems that currently affect several countries around the world in terms of overweight and obesity. These health issues emphasise the need of finding relevant new strategies to face the challenge. In this sense, the elucidation of the mechanisms involved in the browning phenomenon have become an important therapeutic approach. In 2012, Park and Yoon began to evaluate the effect of bioactive compounds from medicinal plants on the mechanisms of adipogenesis and lipid accumulation [28]. These studies, based on bioactive compounds from medicinal plants, established the first evidence of the potential pathways that could explain the phenomenon of browning. In this way, the initial hypothesis was based on AMPK activity mainly due to its relationship with energy metabolism, which has helped to make important advances in the understanding of this process. An important contribution has been the search and identification of

bioactive compounds present in several food sources and their relationship among these and the promotion of browning phenomenon; in this sense, this represents an important objective because new strategies can be proposed based on the diversity and content of these compounds into the diet.

It is important to emphasize differences in the terms used in the literature for referring to this phenomenon of phenotype change within adipose tissue, since browning or transdifferentiation can be used indistinctly regardless of the maturation point of the adipocytes. Therefore, we could refer to transdifferentiation as the process in which a pre-adipocyte, depending on the stimulus, matures directly into a brown or beige adipocyte; meanwhile, browning could start from mature white adipocytes that again, depending on the stimuli, have the ability to modify their phenotype towards beige adipocytes.

The use of cell lines such as 3T3-L1 has allowed us to identify genes related to the browning process (with the most characteristic being PGC-1α, PPARγ, PRMD16 and UCP-1); however, despite the fact that the protocols for the differentiation of this particular cell line are fairly well-described, the concentrations of the pro-differentiating elements as well as the differentiation periods vary among working groups. In addition, other characteristics inherent to the management of this cell line and factors related to its storage and care/handling could interfere with the results obtained so far. As part of the efforts to achieve a better understanding of the mechanisms involved in adipocyte browning, it would be desirable to reach a consensus about the potential problems that could arise when this cell line matures into white adipocytes and the advantages that could arise with respect to primary cell cultures. In this regards, primary cell cultures are currently proposed as more suitable models, despite the fact that they require more rigorous protocols for their care and handling.

Although it has been proposed that inflammatory processes have a close relationship in the progression of overweight and obesity, the role of inflammation as a trigger is currently being explored. Resident macrophages in WAT have become highly relevant in recent years due to their polarization capacity and the effector functions derived from this phenotypic change. In other words, current approaches no longer cover only the processes that occur in adipocytes; instead, a combined strategy is used as an attempt to attack the initiating or promoting mechanisms involved in overweight and obesity by using both cell lines, such as co-culture models consisting in M2 macrophages and adipocytes.

Based on the findings reported in the literature, it is noteworthy to mention that the 3T3-L1 cell line shows several phenotypical characteristics of either WAT and BAT, and to a lesser extent, of BeAT. According to Morrison and McGee [70], the phenotypic variability in this cell line may be due to the presence of specific stimuli. Despite this variability, in addition to the diverse culture media composition used in the thermogenic studies, the use of this model remains widely accepted.

Currently, the use of mesenchymal lines for the study of WAT browning induced by biocompounds from food sources is still scarce, but promising. Therefore, in the near future, it would be expected that these cell lines would result in a better and more widely used model to study these phenomena; however, since an effect similar to that seen with 3T3 cannot yet be ruled out, it is necessary to further explore the effect of bioactive compounds present in food to support the above.

**Author Contributions:** Original draft, writing, review and editing, J.M.-N.; Conceptualization, resources, supervision, writing, review and editing, M.R.-G. All authors have read and agreed to the published version of the manuscript.

**Funding:** This research received no external funding.

**Institutional Review Board Statement:** Not applicable.

**Informed Consent Statement:** Not applicable.

**Data Availability Statement:** Not applicable.

**Conflicts of Interest:** The authors declare no conflict of interest.

## Abbreviations

| | |
|---|---|
| ACOT11 | Acyl CoA thiolesterase. |
| ADBR | Beta adrenergic receptor |
| AHR | Aryl hydrocarbon receptor. |
| AKT | Serine threonine kinase. |
| AMPK | AMP-activated protein kinase. |
| APOL7C | Apolipoprotein L7c. |
| ATCC | American Type Culture Collection. |
| Atg's | Genes related to autophagy. |
| Beclin1 | Beclin 1. |
| BMP7 | Bone morphogenetic protein 7. |
| BMP-8b | Bone morphogenetic protein 8b. |
| C/EBP$\alpha$ | CCAAT alpha enhancer binding protein. |
| CIDEA | Activator of cell death A. |
| COX7A1 | Cytochrome C oxidase 7A1 polypeptide. |
| CPT1B | Carnitine palmitoyltransferase 1. |
| CXCL-14 | C-X-C Motif Chemokine Ligand 14 |
| DAPL1 | Protein associated with apoptosis 1. |
| DIO2 | Iodothyronine deiodinase 2. |
| ELOVL3 | Fatty acid elongase 3. |
| EPDR-1 | Ependymin-related 1. |
| EPSTI1 | Epithelial stromal interaction protein 1. |
| EVA1 | Zero myelin protein type 2. |
| FGF-21 | Fibroblast growth factor 21. |
| GDF-15 | Growth differentiation factor 15. |
| GM-CSF | Macrophage-colony stimulating factor. |
| GRAP2 | GRB2-related adapter protein 2. |
| HOXC8 | Homeobox C8. |
| IGF1 | Insulin growth factor type 1. |
| IGFBP-2 | Insulin-like growth factor binding protein 2. |
| Keap1 | Kelch-type ECH-associated protein 1. |
| LC3II | Autophagosomal marker lysosomal exchanger. |
| LHX8 | LIM homebox 8 protein. |
| MEST | Specific transcript of the mesoderm. |
| mTOR | Mammalian cell rapamycin target. |
| mTORC1 | Mechanistic target of rampamycin complex 1. |
| NF-$\kappa$B | Nuclear factor enhancer of the kappa light chains of activated B cells. |
| Nrf2 | Factor related to erythroid nuclear factor 2. |
| NFG | Nerve growth factor |
| NRG-4 | Neuregulin 4. |
| PI3K | Phosphoinositol 3 kinase. |
| PDK4 | Pyruvate dehydrogenase lipoamide isoenzyme kinase 4. |
| PGC-1$\alpha$ | Peroxisome proliferator activated receptor coactivating protein 1$\alpha$. |
| PLIN1 | Perilipin 1. |
| PPAR-$\gamma$ | Peroxisome proliferator activated gamma receptor. |
| PRDM16 | Histone-lysine N-methyltransferase. |
| Rb | Retinoblastoma protein. |
| RIP140 | Nuclear receptor interaction protein 1. |
| S100B | S100 Calcium Binding Protein B. |
| SCL27A2 | Fatty acid transporter protein 2. |
| SLIT2-c | Slit homolog 2 protein precursor. |
| SNCG | Gamma-synuclein. |
| STAP1 | Signal transduction adapter protein 1. |
| STAT | Signal transducer and activator of transcription protein. |

| TCF21 | Transcription factor 21. |
| TLE3 | Transcriptional corepressor 3. |
| TLR4 | Toll-like receptor 4. |
| TNF-$\alpha$ | Tumor necrosis factor |
| UCP-1 | Uncoupling protein 1. |
| VEGF-A | Vascular endothelial growth factor A |
| ZIC1 | Protein 1 zinc finger. |

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
