# Peer review of "Bioactive Compounds and Adipocyte Browning Phenomenon"

_cimb, doi:10.3390/cimb44070210_

Round 1

Reviewer 1 Report

This article is interesting, compactly summarizing information regarding bioactive compounds that possibly exert metabolism-improving effects by reorganizing the states of adipose tissues including BAT, BeAT and WAT.

This manuscript will be a candidate for the publication in CIMB if the following points are appropriately responded.

Major concerns:

1) In Table 1, information regarding adipokines, including BATokines, should be included since it is known that secreting factors (i.e. hormones and cytokines produced by adipose tissues) play crucial roles in metabolism regulation.

2)Regarding in vitro studies, findings obtained from experiments using human cells (e.g human mesenchymal stroma cells and human iPS cells), not only those using murine 3T3-L1 cells, should be referred.   

Reviewer 2 Report

The manuscript entitled “Bioactive Compounds and Adipocyte Browning Phenomenon”, authored by J Manrique-Nuñez and M Ramos-Gomez, reviews the current knowledge about the effect of different dietary compounds in the browning of adipose tissue, mainly covering those experiments carried out in vitro with 3T3-L1 cells, although they also include in vivo studies. Although the review is well-written and covers an interesting topic, there are few minor questions that authors may consider to improve the review.

-       Gene markers in table 1 should be corrected. For instance, PRDM16, PPARGC1A, CIDEA, COX7A1, EPSTI1 and LHX8 have been described to be expressed in both brown and beige adipocytes. Thus, these markers are not exclusively informative of only one type of adipocyte. The same occurs with HOXC8 because, although it has been described to be negatively correlated with UCP1 expression, it is expressed in both white and beige adipose tissue. The references included to support this information [4, 10, 13] are out of date. Authors should consult more recent bibliography (i.e. PMID 30332656 and 33776911).

-       In the same table 1, I would use the term “Primary function” instead of “function”, because it’s true that energy storing or heat production are WAT and BAT functions, but they also exert other functions, such as endocrine function.

-       3T3-L1 adipocytes should not be considered as “pure” white adipocytes because their capacity to increase oxygen consumption after treatment with catecholamines is UCP1-dependent, among other characteristics closer to brown adipocytes, like multilocular lipid droplets (for more information, PMID 26451286). Authors should not assume that 3T3-L1 cells differentiate into white adipocytes but into beige adipocytes, and probably, the effects of the biocompounds tested on these cells are to accelerate/enhance browning. Authors should include this information as a limitation of the studies using 3T3-L1 cells.

-       In section 6, according to the title, readers expect to find studies focused on the capacity of dietary compounds to promote browning through the polarization of macrophages. However, only reference [60] studies the effect of curcumin in this scenario, and the compound isn’t even mentioned. Authors should expand this section including references that explicitly study the effects of biocompounds on macrophage polarization during browning.

Round 2

Reviewer 1 Report

In the revised manuscript, authors have fully responded the comments. The reviewer thinks that the revised manuscript is a strong candidate for publication in CIMB.

This manuscript is a resubmission of an earlier submission. The following is a list of the peer review reports and author responses from that submission.